# Electronic, Optical, Mechanical, and Electronic Transport Properties of SrCu_2_O_2_: A First-Principles Study

**DOI:** 10.3390/ma16051829

**Published:** 2023-02-23

**Authors:** Sheng Jiang, Chaohao Hu, Dianhui Wang, Yan Zhong, Chengying Tang

**Affiliations:** 1School of Materials Science and Engineering, Guilin University of Electronic Technology, Guilin 541004, China; 2Guangxi Key Laboratory of Information Materials, Guilin University of Electronic Technology, Guilin 541004, China

**Keywords:** SrCu_2_O_2_, optical properties, carrier mobility, mechanical properties, first-principles calculations

## Abstract

The structural, electronic, optical, mechanical, lattice dynamics, and electronic transport properties of SrCu_2_O_2_ crystals were studied using first-principles calculations. The calculated band gap of SrCu_2_O_2_ using the HSE hybrid functional is about 3.33 eV, which is well consistent with the experimental value. The calculated optical parameters show a relatively strong response to the visible light region for SrCu_2_O_2_. The calculated elastic constants and phonon dispersion indicate that SrCu_2_O_2_ has strong stability in mechanical and lattice dynamics. The deep analysis of calculated mobilities of electrons and holes with their effective masses proves the high separation and low recombination efficiency of photoinduced carriers in SrCu_2_O_2_.

## 1. Introduction

In response to the problems of energy shortage and environmental pollution, photocatalytic technology, which converts pollutants into electrical and chemical energy with suitable catalysts under light, has gradually developed [1]. In 1972, Fujishima et al. [2] discovered the phenomenon of photocatalytic water decomposing on TiO_2_ electrodes, which not only opened up a new field of semiconductor photocatalysis, but also made semiconductor materials occupy an essential role in photocatalysis with organic material [3]. However, most of the reported photocatalysts are wide-bandgap semiconductors with little response to visible light, which limits their application in the field of photocatalysis. Therefore, except for proper elemental doping [4], the search for photocatalytic semiconductors with high utilization of visible light has become the focus of research in recent years.

Recently, transparent conductive oxides (TCO), most of which are semiconductors, have attracted extensive attention due to their good electrical conductivity [5]. Among them, Cu_2_O is a typical p-type semiconductor and has been widely studied since it has a suitable band gap (2.0–2.2 eV), is environmentally friendly, and has low toxicity [6]. At present, it is also used as an important photocatalyst with high visible light utilization in the field of hydrogen hydrolysis and organic pollutant degradation [7,8,9]. The valence band characteristics of Cu_2_O are mainly ascribed to the special O-Cu-O dumbbell structure [10,11], which makes us convinced that other monovalent copper-based oxides with an O-Cu-O dumbbell structure also maintain similar valence band properties and physicochemical properties. Actually, the SrCu_2_O_2_ considered in this work belongs to one kind of monovalent copper-based oxides and also possesses the O-Cu-O dumbbell-like structural units. Generally, SrCu_2_O_2_ is often reported as one kind of TCO used in thin film coating fields [12,13,14]. According to the best of our knowledge, however, the potential application of SrCu_2_O_2_ in the field of photocatalysis has not attracted much attention up to now, and the investigations on its physical properties especially involving its optical, mechanical, and electron transport properties are still relatively scarce.

In recent years, as first-principles calculations have become the main guiding method for material theory research, it has been a reality to accurately predict the ground state properties of some new materials [15,16]. Furthermore, the measurement of = transport properties is more difficult than the properties related to the total energy, and there is a lack of sufficient reference data to verify them [17]. Thus, the predictions of properties involving electronic excitation are still less mature. For example, a combination of ab initio calculations and semi-empirical formulations is usually used to predict the thermoelectric transfer coefficient [18,19,20]. The carrier mobility *μ* is closely related to electronic excitation. As an important parameter to measure the conductive properties of semiconductors, it quantifies how fast the electrons or holes can travel inside the semiconductor under external electric field *E*. Thus, it is necessary to predict the carrier mobility accurately. The well-known Drude formula for electron mobility *μ = eτ/m** indicates that an accurate carrier mobility prediction requires an accurate calculation of the scattering rate *1/τ* and the carrier effective mass *m**.

At present, the theoretical calculation of SrCu_2_O_2_ is basically limited to the calculation of electronic properties. Ohta [21] calculated the band structure of SrCu_2_O_2_ using the density functional theory (DFT) with the local density approximation (LDA) function and obtained a band gap of about 1.83 eV; S. Boudin et al. [22] calculated an about 2.0 eV band gap of SrCu_2_O_2_ using atomic sphere approximation. In the present work, we investigated the structural, electronic, optical, mechanical, lattice dynamics, and electron transport properties of perfect SrCu_2_O_2_ crystals using first-principles calculations. We believe that these detailed calculated results will provide a good guide for future research on SrCu_2_O_2_.

## 2. Computational Details

Except for the calculations of electronic transport properties, other electronic structure calculations were implemented in VASP code [23,24], with the projector augmented wave (PAW) method [25,26] used to describe the ion–electron interaction. While the exchange correlation potential between electrons is the generalized gradient approximation (GGA) function modified by Perdew, Burke, and Ernzerhof (PBE) [27]. In addition, the cut-off energy of the plane wave is set to 420 eV. In the process of full relaxation, the Brillouin zone is sampled by a 4 × 4 × 4 gamma-centered k-point grid. The total energy and force on each atom are set to converge to 1.0 × 10^−5^ eV/atom and 0.01 eV/Å in the self-consistent cycle process to ensure that the system reaches the approximate most stable state. Considering that the DFT method may lead to underestimation of the band gap, the Heyd–Scuseria–Ernzerhof (HSE) [28,29] hybrid functional (including 25% precise exchange, 75% PBE exchange, and 100% PBE correlation energy in this work) was also used in electronic structure calculations. During the calculations on the optical properties, a denser 6 × 6 × 6 k-point sampling in the Brillouin zone was adopted to improve the calculation accuracy, while 52 additional empty bands were added to maintain the excited states. An 8 × 8 × 8 gamma-centered k-point grid was used to calculate the mechanical properties as it requires a sufficiently large k-point density. Based on density functional perturbation theory (DFPT) [30,31], phonon calculations were performed to investigate the lattice dynamical properties of SrCu_2_O_2_. With a 2 × 2 × 2 supercell of SrCu_2_O_2_, the phonon dispersion curve and phonon density of states were calculated using the PHONOPY code [32], which extracts the force constant obtained by the VASP code.

For investigating the electronic transport properties of SrCu_2_O_2_, EPW code [33,34] under Quantum Espresso software (version 7.1) [35,36] was used with the PBE functional within GGA. All the pseudopotentials are projector augmented wave (PAW) types generated by A. Dal Corso using an “atomic” code. In addition, the kinetic energy cutoffs for wavefunctions and for charge density and potential are set to 100 Ry and 350 Ry, respectively. In the relaxation process, the interatomic force and pressure converged to 1 × 10^−3^ a.u and 0.01 GPa, respectively, while the relative threshold of self-consistent convereged to 1 × 10^−13^ a.u. Meanwhile, the Brillouin zone was sampled by an 8 × 8 × 8 gamma-centered k-point to obtain convergent linear response quantities such as dielectric tensor using the DFPT method. Finally, to converge carrier mobility, the EPW code was used with the maximum localized Wannier functions (MLWF) [37] in WANNIER90 software (version 3.1.0) [38] to interpolate the electron–phonon matrix elements from a coarse 8 × 8 × 8 k-point and a 2 × 2 × 2 q-point grid to a fine 40 × 40 × 40 k and q-point grid. To reduce computational costs, only those matrix elements within 0.948090 eV of the Fermi surface states were chosen to calculate carrier mobilities. Moreover, in this work, the self-consistent iteration (IBTE) and the self-energy relaxation time approximation (SERTA) were used to solve the Boltzmann transport equation (BTE).

## 3. Results and Discussion

### 3.1. Crystal Structure of SrCu_2_O_2_

At ground state, the SrCu_2_O_2_ shown in Figure 1 has a body-centered tetragonal crystal cell with a I41/amd space group. Sr atoms are located in the center of the octahedrons and are composed of 6 O atoms, while the Cu atoms and the O atoms form a special O-Cu-O dumbbell-like structure. The optimized lattice constants listed in Table 1 agree well with the experimental values [39].

### 3.2. Electronic Properties of SrCu_2_O_2_

The band structure along the high symmetry points in the Brillouin zone was calculated and is depicted in Figure 2. The band structure calculated by the traditional DFT method (the dashed line in the figure) shows that the band gap of SrCu_2_O_2_ is about 1.82 eV, which is far from the experimental value [40]. It is well known that the traditional DFT calculations ignore the coulomb interaction between the excited electrons and usually lead to underestimation of the band gap. The corrected band structure (the solid line in Figure 2) calculated based on the HSE hybrid functional shows that SrCu_2_O_2_ is a semiconductor with a direct band gap of 3.33 eV at the Γ point, which agrees perfectly with the experimental value (3.3–3.35 eV) [40,41]. Furthermore, the band dispersion at the top of the valence band is relatively flat, while the band at the bottom of the conduction band shows a strong dispersion relationship along the Γ-Z path, reflecting the mixing of light and heavy bands.

The effective mass (*m**) can be further obtained from the calculated energy band structure and *m** is defined as follows:(1)$${\left(\frac{1}{m^{*}}\right)}_{ij}=\frac{1}{\hbar^{2}}\frac{\partial^{2}E\left(k\right)}{\partial k_{i}k_{j}},i,j=x,y,z\;$$
where *k* is the wave vector, *E*(*k*) is the corresponding band energy, and ℏ expresses the reduced Planck constant. The calculated effective mass of holes (*m*_h_*) and electrons (*m*_e_*) of SrCu_2_O_2_ at the *Γ* point near the Fermi level are shown in Table 2. It can be found that the effective mass of carriers in SrCu_2_O_2_ shows obvious anisotropy. The values of *m*_e_*/*m*_h_* in the *x* and *y* directions are 4.7 and the value of *m*_h_*/*m*_e_* in the *z* direction is about 5.9, which are much larger than the relative value of the widely used photocatalytic material anatase TiO_2_ (2.1) [42]. Generally, the values of *m*_h_* and *m*_e_* and the difference between them directly reflect the mobility of electrons and holes and the corresponding separation efficiency to some extent. Thus, it can be considered that SrCu_2_O_2_ is a semiconductor photocatalyst with higher separation efficiency and a low recombination rate of photogenerated electrons and holes in comparison with TiO_2_.

To study the electronic properties of SrCu_2_O_2_ more deeply, the corresponding electronic total and partial density of states within the HSE hybrid functional calculations were further investigated and are shown in Figure 3. The electronic states near to the top of the valence band are mainly contributed to by Cu-3d and O-2p electrons. The sharp peak indicates that these electrons are localized at these energy levels. The states around the bottom of the conduction band are mainly composed of Sr-4d and Cu-3d, as well as O-2p electrons. What is more, the strong hybridization between Sr-4d and O-2p in the SrO_6_ octahedron and the bonding interactions of Cu-3d and O-2p in the O-Cu-O dumbbell-like structural units could be considered as the essential reasons for the structural stability of the SrCu_2_O_2_ system.

To further explore the bonding information between Sr, Cu, and O atoms in SrCu_2_O_2_, the electron localization function (ELF) was calculated and is shown in Figure 4. The values of the ELF are admittedly between 0 and 1. According to its original definition, the bonding interactions between atoms are obviously covalent if the ELF is close to 1. The interactions between atoms correspond to ionic bonds while the value of the ELF is about zero. The ELF between Sr and O is close to zero, indicating the typical ionic bonding. While the ELF between Cu and O is about 0.2, which shows that the ionic interaction between Cu and O atoms in the O-Cu-O structure is primary.

### 3.3. Optical Properties of SrCu_2_O_2_

To explore the optical properties of SrCu_2_O_2_, its dielectric function with respect to frequency was calculated based on the band structures obtained by the HSE hybrid functional method. The calculated dielectric function including its real part and imaginary part along the (100) and (001) directions is shown in Figure 5. The anisotropy of dielectric function is in consistent with the crystal structure of SrCu_2_O_2_. The static dielectric constant is 6.29 and 5.92 along (100) and (001) directions, respectively. With the increase of photon energy, the real part *ε*_1_ increases gradually and reaches a maximum value of 10.34 at 2.41 eV along the (100) direction and reaches 9.77 at 3.14 eV along the (001) direction. When the photon energy reaches the electronic band gap of 3.32 eV, the electrons in the valence will be excited and transit to the conduction band, while the holes formed subsequently stay in the valence band. With further enhancement of the photon energy, the dielectric function produces numerical fluctuations corresponding to the carrier concentration. The imaginary part *ε*_2_ can be regarded as the properties of the light absorption behavior. The peak of *ε*_2_ appears around 2.6~3.8 eV in all directions, which may be caused by the transition of electrons from the top of the valence band to the bottom of the conduction band.

The complex refractive index *n*, extinction coefficient *k*, absorption coefficient *α*, reflectance *R,* and energy loss spectrum *L* of SrCu_2_O_2_ in all directions were further calculated and are shown in Figure 6. There are high peaks of *n* in the region of 1.5~3.3 eV, which corresponds to the wavelength range of 376~828 nm and shows that SrCu_2_O_2_ has high refractivity of visible light. Further analyzing the data in the visible light region (1.5–3.3 eV) enlarged in the insets of Figure 6, the complex refractive index *n* reaches a maximum value of 3.26 at 2.41 eV along the (100) direction and reaches 3.22 at 3.17 eV along the (001) direction. The value of extinction coefficient *k* increases sharply from 1.5 eV, also reflecting the response of SrCu_2_O_2_ to visible light. For the calculated absorption coefficient *α* presented in Figure 6c, the first obvious peak representing the absorption edge is close to 3.32 eV, which is consistent with the electronic band gap calculated using the HSE hybrid functional method and also agrees with the experimental value [40,41]. Moreover, the strongest absorption peaks along the (100) and (001) directions are around 23 eV and 26 eV, respectively. The reflectance *R* reflects the resonant frequency of the incident light. It can be found from Figure 6d that the average value of *R* is about 9.5%, indicating that SrCu_2_O_2_ can be regarded as a good light-absorbing material. The energy loss spectrum *L* represents the resonant frequency of the plasma. It can be seen from Figure 6e that the peak of *L* is near 29 eV, and the position of the energy loss peak corresponds to the position where the reflection spectrum drops sharply. Meanwhile, the calculated *L* is completely within the continuous energy range of 0~55 eV, indicating an unobvious plasma oscillation of SrCu_2_O_2_.

### 3.4. Electronic Transport Properties of SrCu_2_O_2_

To investigate the electronic transport properties of SrCu_2_O_2_, its internal ab initio drift carrier mobility was calculated. The self-consistent iteration (IBTE) and self-energy time relaxation approximation were used (SERTA) to solve the Boltzmann transport equation (BTE). The electron and hole drift mobility tensors at room temperature obtained using SERTA are listed in Table 3.

The calculated drift carrier mobility of SrCu_2_O_2_ as a function of temperature is shown in Figure 7, with the carrier concentrations all set to 10^−13^ cm^−3^. At a low temperature, the carrier mobility obtained by IBTE slightly differs from SERTA. However, as the temperature increases, the calculated values tend to be consistent. At room temperature, the average of the electron and hole drift mobilities obtained by IBTE are 80.6 cm^2^/Vs and 31.3 cm^2^/Vs, respectively, while those obtained by SERTA are 84.5 cm^2^/Vs and 33.0 cm^2^/Vs, respectively. Moreover, the mobility of electrons along the (001) direction is much higher than that in the (100) and (010) directions, while the situation of the holes is exactly the opposite, which is consistent with the results of effective masses. The ratios of average mobility of electrons to holes $\bar{\mu_{e}}/\bar{\mu_{h}}$ are always greater than 2, which explains the high separation efficiency and low recombination rate for the photogenerated electrons and holes.

### 3.5. Lattice Dynamics Properties of SrCu_2_O_2_

To further study the lattice dynamics of SrCu_2_O_2_, the phonon dispersion curve along high symmetry directions and the corresponding phonon density of states were calculated as shown in Figure 8. There are 20 atoms in a unit cell of SrCu_2_O_2_, thus there are 60 branches in the phonon spectrum, including 3 acoustic branches and 57 optical branches. There is no imaginary frequency in the calculated phonon spectrum, showing the good dynamic stability of SrCu_2_O_2_. The acoustic branches reflect the vibration of the primitive cell’s centroid and occupy the 0~2.8 THz frequency region. Furthermore, for the acoustic branches through *Γ*, the frequency of the longitudinal modes is higher than that of the transverse modes. In addition, there is no separation between the longitudinal acoustic branch and the transverse optical branch around 2.8 THz, indicating that the transition from the acoustic mode to the optical mode does not require any momentum transfer [43]. Combined with the phonon density of states, the coupling vibration of Sr, Cu, and O atoms is mainly manifested below 8.6 THz, especially in the frequency range of 1.2~7.0 THz, which shows a difference between the coupling vibration of Sr-O and Cu-O. In the range of 14.4~16.6 THz, the phonon branch is basically attributed to the vibration of O atoms, while it is contributed to by the coupled vibrations of Cu and O atoms for 17.0~19.4 THz.

### 3.6. Mechanical Properties of SrCu_2_O_2_

To study the mechanical properties of SrCu_2_O_2_, the elastic constants *C*_ij_ of SrCu_2_O_2_ were calculated using stress-strain method [44], and the results are listed in Table 4. For tetragonal crystals, the mechanical stability criterion [45] is given by the following formula:(2)$$\left(C_{11}-C_{12}\right)>0,\left(C_{11}+C_{33}-2C_{13}\right)>0,\;C_{11}>0,C_{33}>0,C_{44}>0,C_{66}>0\;\left(2C_{11}+2C_{12}+4C_{13}+C_{33}\right)>0$$

The calculated *C*_ij_ satisfies the above relationship, indicating the mechanical stability of the SrCu_2_O_2_ crystal. *C*_11_ is significantly smaller than *C*_33_, indicating that the chemical bond strength of SrCu_2_O_2_ along the (100) and (010) directions is significantly weaker than that along the (001) direction. At the same time, *C*_44_ is slightly larger than *C*_66_, which proves that the (001) direction is less prone to shear deformation than the (010) direction.

For the tetragonal crystal, according to the Voigt–Reuss–Hill [46,47,48,49] theory, the Voigt bulk modulus *B_V_* and shear modulus *G_V_* with Reuss bulk modulus *B_R_* and shear modulus *G_R_* of SrCu_2_O_2_ can be calculated by the following formula:(3)$$B_{V}=\left(2C_{11}+2C_{12}+C_{33}+4C_{13}\right)/9\;$$
(4)$$B_{R}=C^{2}/M$$
(5)$$G_{V}=\left(M+3C_{11}-3C_{12}+12C_{44}+6C_{66}\right)/30$$
(6)$$G_{R}=15\left\{\left(18B_{V}/C^{2}\right)+\left[6/\left(C_{11}-C_{12}\right)\right]+\left(6/C_{44}\right)+\left(3/C_{66}\right)\right\}{\;}^{-1}$$
(7)$$M=C_{11}+C_{12}+2C_{33}-4C_{13}$$
(8)$$C^{2}=\left(C_{11}+C_{12}\right)C_{33}-2C_{13}^{2}$$

According to the Hill model, the calculated values of the Voigt model represent the maximum of the elastic modulus, while the Reuss model signifies the minimum, and the arithmetic mean of the Reuss and Voigt models is taken as the bulk modulus *B* and the shear modulus *G* shown in Table 3, expressed as:(9)$$B=B_{H}=\left(B_{V}+B_{R}\right)/2$$
(10)$$G=G_{H}=\left(G_{V}+G_{R}\right)/2$$

In addition, Young’s modulus *E*, Poisson’s ratio *ν*, and Cauchy pressure *C*′ can be calculated respectively by the following formulas:(11)$$E=9BG/\left(3B+G\right)$$
(12)$$\nu =\left(3B-2G\right)/\left(6B+2G\right)$$
(13)$$C^{\prime }=\left(C_{12}-C_{44}\right)/2$$

It can be clearly seen from Table 3 that the *B*/*G* value is 2.79, which is slightly larger than 1.75. According to Pugh’s criterion [44], this indicates that SrCu_2_O_2_ has good toughness, which is consistent with Poisson’s ratio v and Cauchy pressure *C*′ (*v* > 1/3, *C*′ > 0).

The anisotropy of shear elastic can be described by *A* = 2*C*_66_/(*C*_11_ − *C*_12_). Generally, *A* = 1 means that the material is elastically isotropic, and the farther the value of *A* is from 1, the more obvious the elastic anisotropy of the material. The calculated *A* of SrCu_2_O_2_ is 0.66, indicating that SrCu_2_O_2_ has significant elastic anisotropy. Moreover, the direction-dependent Young’s modulus directly determines the elastic anisotropy of SrCu_2_O_2_. The calculated direction-dependent Young’s modulus is shown in Figure 9. The elasticity of SrCu_2_O_2_ is relatively uniform in the *x* and *y* directions, while its anisotropy is mainly reflected in the *z* direction, whose value is about twice that in the *x* and *y* directions, which is consistent with the lattice parameters of SrCu_2_O_2_.

## 4. Conclusions

In this work, based on first-principles calculations, the structural, electronic, optical, mechanical, lattice dynamics, and electron transport properties of perfect SrCu_2_O_2_ crystals were investigated in depth. The calculated band gap of SrCu_2_O_2_ using the Heyd–Scuseria–Ernzerhof hybrid functional is about 3.33 eV, with it being identical to the experimental value (3.3 eV). The calculated effective masses reveal a huge difference between the mobilities of electrons and holes in all directions, indicating the low recombination rate in SrCu_2_O_2_. Meanwhile, the calculated optical parameters demonstrate a relatively strong response to visible light for SrCu_2_O_2_. The calculated elastic constants and phonon dispersion indicate that SrCu_2_O_2_ has good mechanical and lattice dynamics stability. Furthermore, the calculated average mobilities of drift electrons and holes in SrCu_2_O_2_ at room temperature using IBTE are 80.6 cm^2^/Vs and 31.3 cm^2^/Vs, respectively, while those obtained by SERTA are 84.5 cm^2^/Vs and 33.0 cm^2^/Vs, respectively, which is consistent with the calculated effective masses. Moreover, the anisotropy of SrCu_2_O_2_ as its tetragonal system characteristics is reflected through the calculations.

## Figures and Tables

**Figure 1 materials-16-01829-f001:**
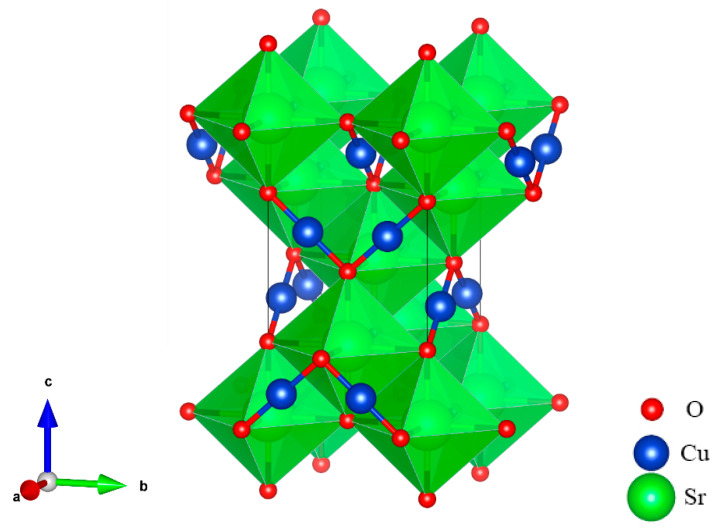
Crystal structure of SrCu_2_O_2_ with the space group I41/amd.

**Figure 2 materials-16-01829-f002:**
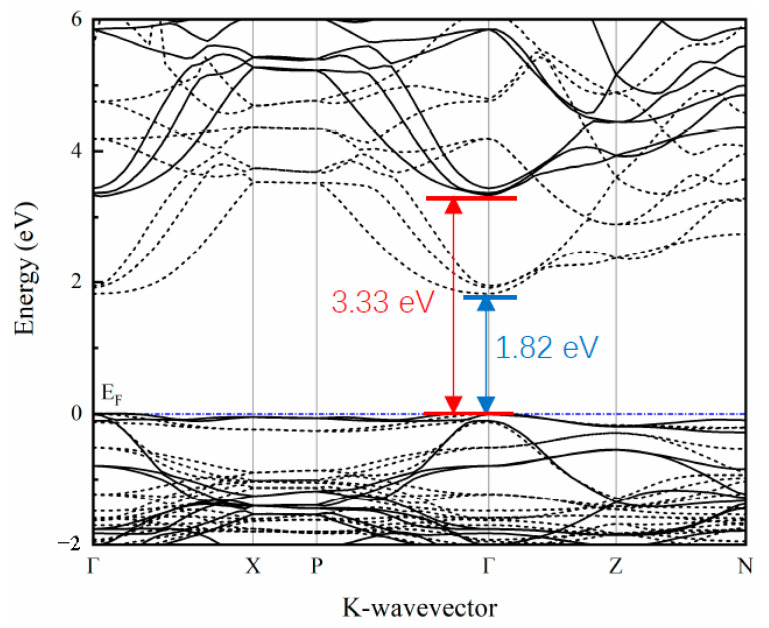
Calculated band structure of SrCu_2_O_2_ using the HSE hybrid functional method (solid line). That band structure calculated with traditional the DFT method (dashed line) is also shown for comparison.

**Figure 3 materials-16-01829-f003:**
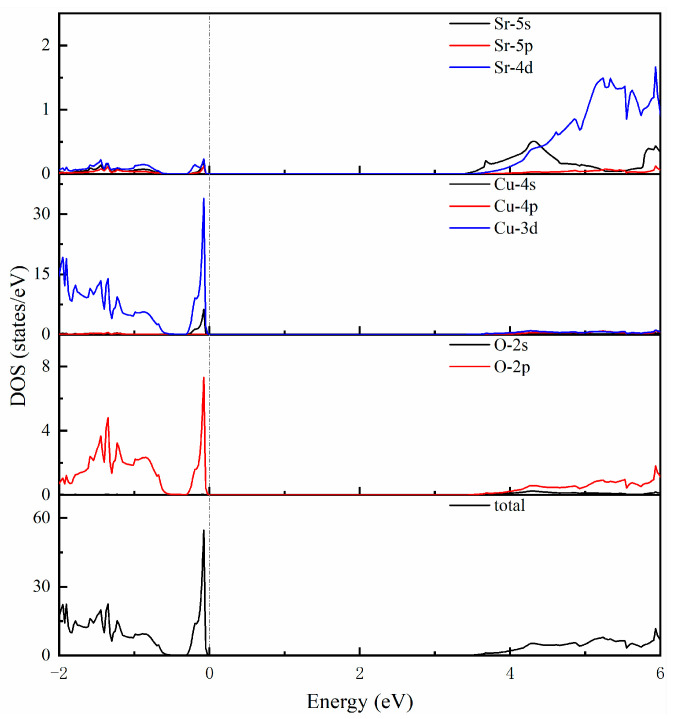
Calculated total and partial density of states using the HSE hybrid functional method.

**Figure 4 materials-16-01829-f004:**
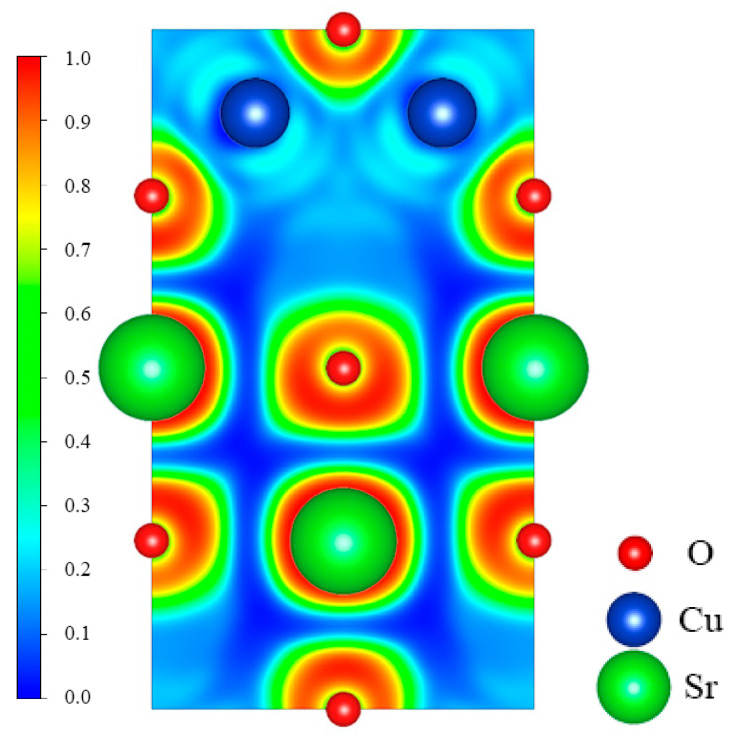
Calculated electron localization function (ELF) for the (010) plane across Sr, Cu, and O atoms.

**Figure 5 materials-16-01829-f005:**
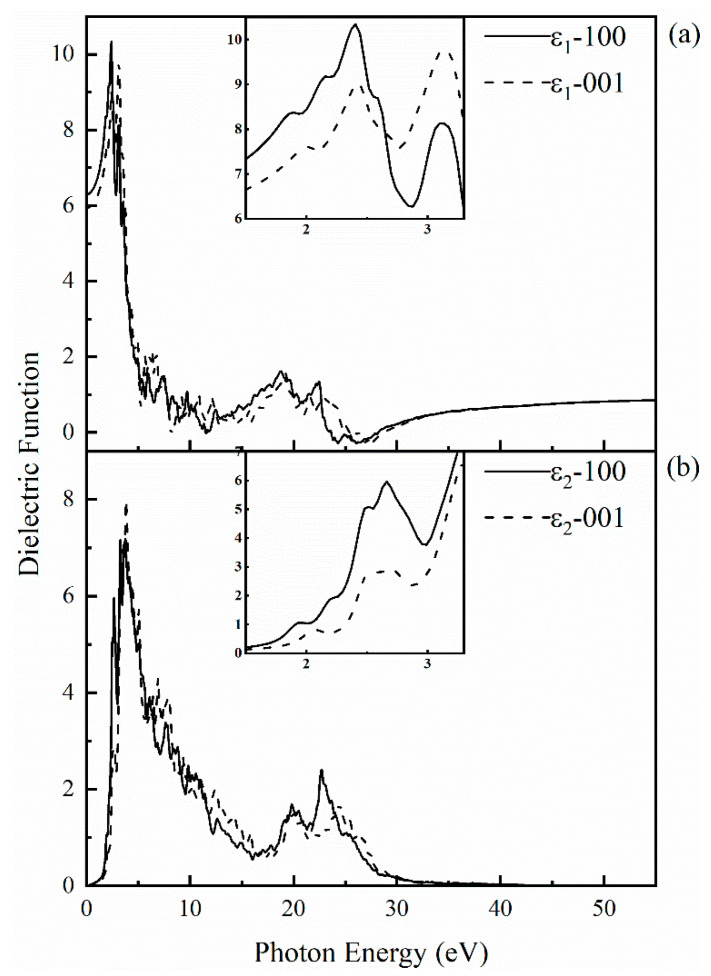
Calculated complex dielectric function for SrCu_2_O_2_: (**a**) the real part along the (100) and (001) directions; (**b**) the imaginary part along the (100) and (001) directions. The insets present the obtained data for the real and imaginary parts of the dielectric function in the visible light region (1.5–3.3 eV), respectively.

**Figure 6 materials-16-01829-f006:**
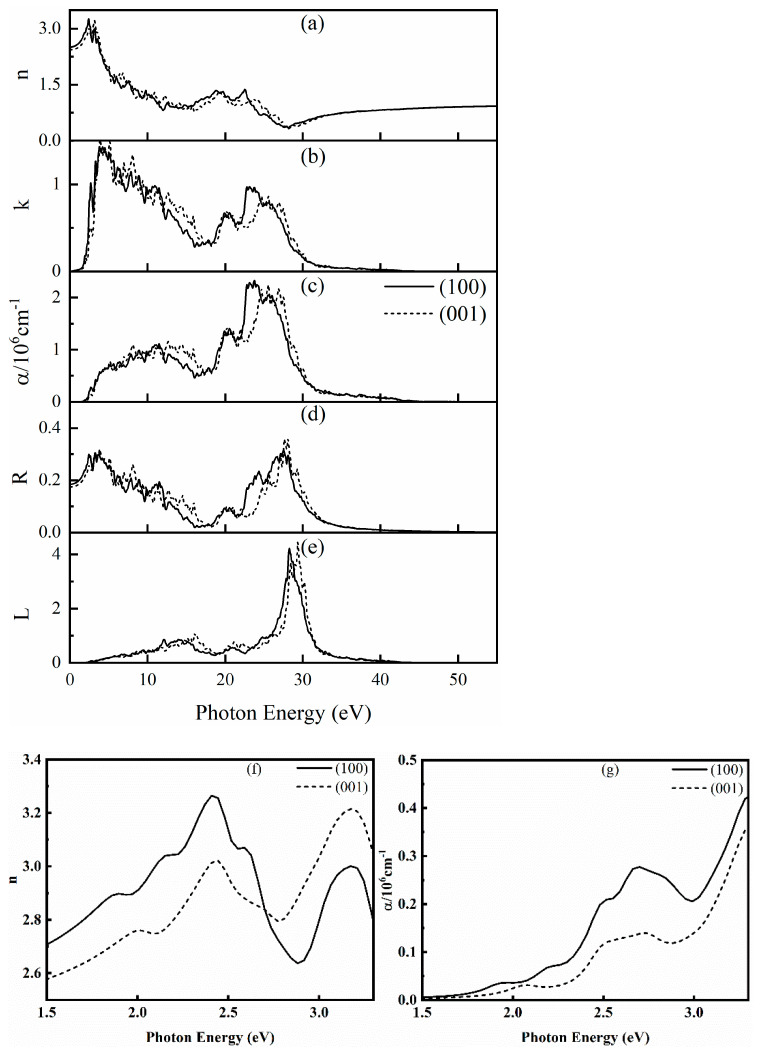
Calculated optical properties of SrCu_2_O_2_: (**a**) complex refractive index *n*, (**b**) extinction coefficient *k*, (**c**) absorption coefficient *α*, (**d**) reflection rate *R*, (**e**) energy loss spectrum *L*, (**f**,**g**) enlarged insets for the complex refractive index *n* and absorption coefficient *α* in the visible light region (1.5–3.3 eV), respectively.

**Figure 7 materials-16-01829-f007:**
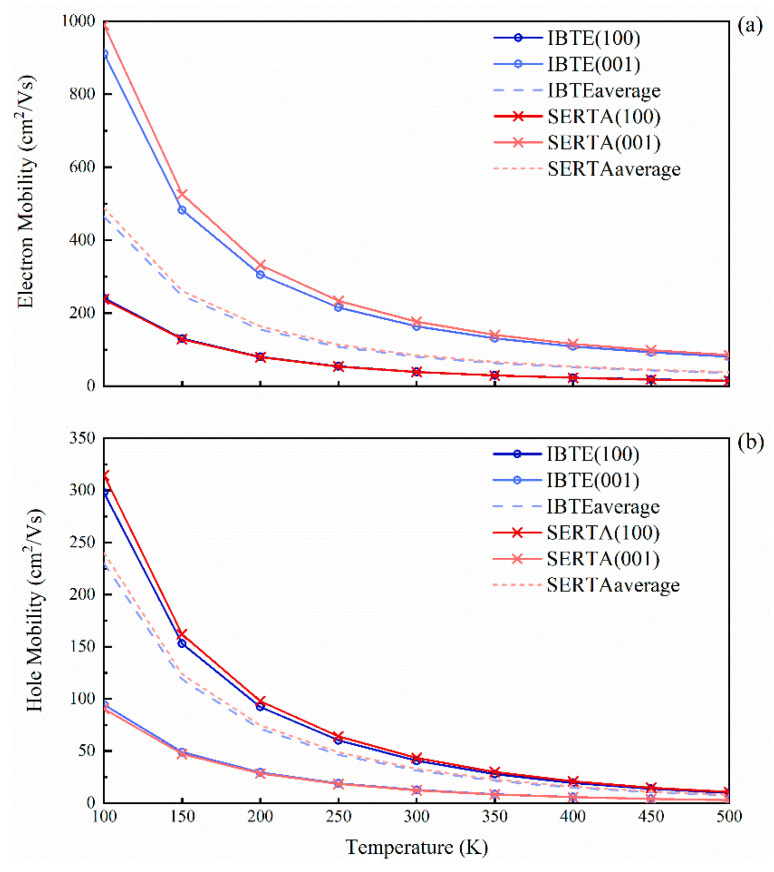
Calculated carrier drift mobilities of SrCu_2_O_2_ as a function of temperature: (**a**) electron mobilities and (**b**) hole mobilities. The line with “o” represents using IBTE, and the line with “x” represents using SERTA, while the dashed line indicates the average values of the mobilities.

**Figure 8 materials-16-01829-f008:**
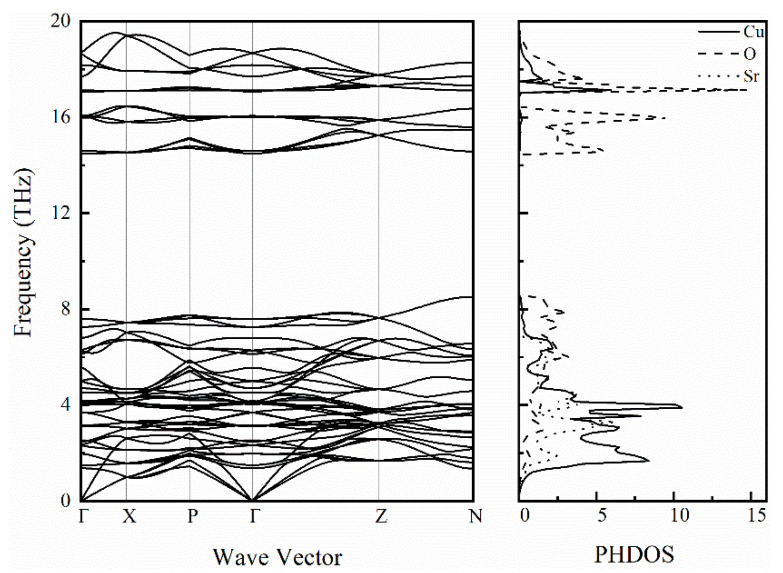
Calculated phonon spectrum and the corresponding phonon density of states for SrCu_2_O_2_.

**Figure 9 materials-16-01829-f009:**
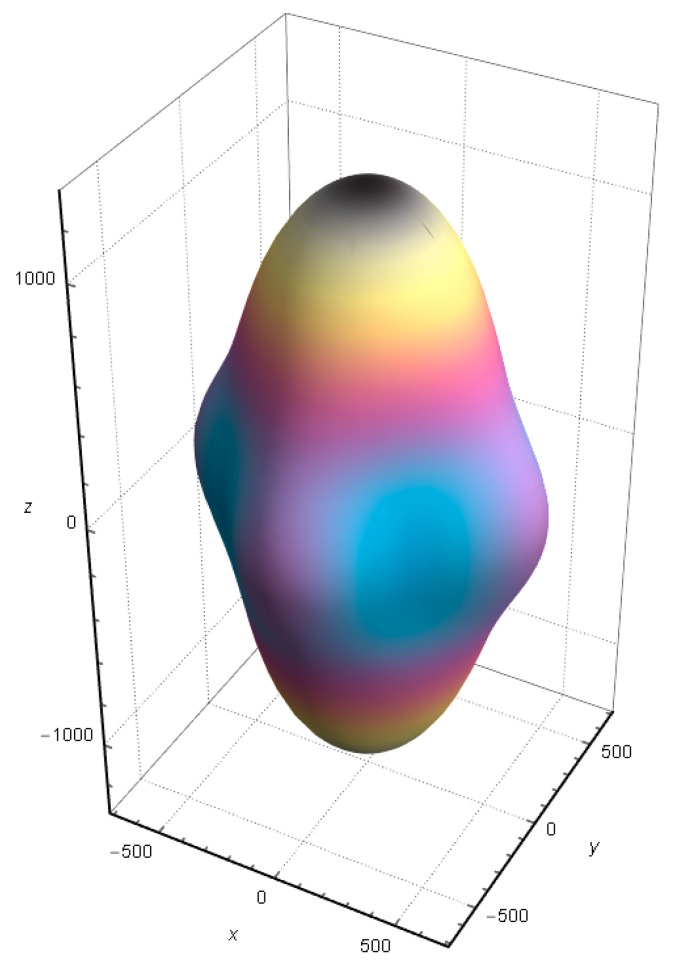
Calculated direction-dependent Young’s modulus of SrCu_2_O_2_.

**Table 1 materials-16-01829-t001:** Optimized structural parameters of SrCu_2_O_2_. The corresponding experimental values [39] are given in the following parentheses for comparison.

Lattice Constants	Atomic Coordinates (Fractional)				
(Å)	Atom	Site	x	y	z
a = 5.5292 (5.48)	Cu	8d	0.375	0.125	0.75
c = 9.8272 (9.82)	O	8e	0.7504 (0.75)	0.7504 (0.75)	0
	Sr	4a	0	0	0

**Table 2 materials-16-01829-t002:** Calculated effective mass of electron (*m*_e_*) and that of hole (*m*_h_*) in three principal directions at the *Γ* point. All values are in units of free electron mass (*m*_0_).

	Conduction Band	Valence Band
*Γ*	$\left(\begin{array}{c} 4.172\;\;\;\;\;\;\;\;\;\;0\;\;\;\;\;\;\;\;\;\;\;\;\;0\;\;\;\;\\ 0\;\;\;\;\;\;\;\;\;4.172\;\;\;\;\;\;\;\;\;\;0\\ \;\;\;\;\;0\;\;\;\;\;\;\;\;\;\;\;\;\;\;\;0\;\;\;\;\;\;\;\;\;\;0.564 \end{array}\right)$	$\left(\begin{array}{c} -0.912\;\;\;\;\;\;\;\;\;\;0\;\;\;\;\;\;\;\;\;\;\;\;\;0\;\;\;\;\\ 0\;\;\;\;\;\;\;-0.912\;\;\;\;\;\;\;\;\;\;0\\ \;\;\;\;\;0\;\;\;\;\;\;\;\;\;\;\;\;\;\;\;0\;\;\;\;\;\;\;-3.316 \end{array}\right)$

**Table 3 materials-16-01829-t003:** Calculated mobility tensors of electrons (*µ_e_**) and holes (*µ_h_**) using the SERTA and IBTE methods.

	*μ_e_*	*μ_h_*
SERTA	$\;\left[\begin{array}{ccc} 38.6 & 0 & 0\\ 0 & 38.6 & 0\\ 0 & 0 & 176.4 \end{array}\right]$	$\left[\begin{array}{ccc} 43.4 & 0 & 0\\ 0 & 43.4 & 0\\ 0 & 0 & 12.3 \end{array}\right]$
IBTE	$\left[\begin{array}{ccc} 39.1 & 0 & 0\\ 0 & 39.1 & 0\\ 0 & 0 & 163.7 \end{array}\right]$	$\left[\begin{array}{ccc} 40.7 & 0 & 0\\ 0 & 40.7 & 0\\ 0 & 0 & 12.7 \end{array}\right]$

**Table 4 materials-16-01829-t004:** Calculated elastic constants *C*_ij_, bulk modulus *B*, shear modulus *G*, Young’s modulus *E*, Poisson’s ratio *ν*, Pugh’s empirical parameter *B*/*G,* and Cauchy pressure *C*′.

Elastic Constants (GPa)						Elastic Moduli (GPa)					
$C_{11}$	$C_{12}$	$C_{13}$	$C_{33}$	$C_{44}$	$C_{66}$	B	G	E	ν	B/G	$C^{\prime }$
94.9	35.6	68.3	192.6	23.6	19.5	72.9	26.2	70.1	0.34	2.79	6.0

## Data Availability

The data presented in this study are available upon request from the corresponding author.

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
