# Peer review of "Electronic, Optical, Mechanical, and Electronic Transport Properties of SrCu2O2: A First-Principles Study"

_materials, 2023, doi:10.3390/ma16051829_

Round 1
Reviewer 1 Report
This paper investigates various electronic, optical, structural properties of SCO using an improved first principles approach of the HSE hybrid functional method. Previous studies were limited due to limitations in DFT. By doing so the authors claim that they can better estimate of various material properties. Overall, the topic is of interest in the community and the author appears to use a reasonably good scientific approach in their investigation. However, I would request further improvements in the manuscript before it can be considered for publication. The authors are encouraged to address the following:
1. The introduction is very hard to follow and does not sufficiently explain the importance of photocatalytic semiconductors and why SCO is a good candidate for investigation. The authors should also include potential applications and overall benefits to studying this material. Please provide background to any jargon so that people less familiar with this field can follow along.
2. For page 2 line 47, can you elaborate more as to what you mean by electronic excitation. Do you mean application of an E-field?
3. For page 4 line 138-140, can you elaborate more on why you conclude that SCO has higher separation efficiency and low recomb rate of photogenerated electrons and holes compared to TiO2.
Author Response
Thanks very much for your suggestions about our manuscript. We have made revisions to the manuscript according to the comments, and point-to-point replies are listed as follows:
- The introduction is very hard to follow and does not sufficiently explain the importance of photocatalytic semiconductors and why SCO is a good candidate for investigation. The authors should also include potential applications and overall benefits to studying this material. Please provide background to any jargon so that people less familiar with this field can follow along.
Reply: We have added a more in-depth description of the necessity and feasibility of the research about SrCu2O2 in the introduction. These can be seen on page 1, line 23-31, line 35-48 and page 2, line 69-71 of the revised manuscript.
- For page 2 line 47, can you elaborate more as to what you mean by electronic excitation. Do you mean application of an E-field?
Reply: For semiconductors, electrons need to get energy from elsewhere like photon, E-field, force etc. to gain transport properties. Electronic excitation we mentioned here means the transition from the ground state to the excited state.
- For page 4 line 138-140, can you elaborate more on why you conclude that SCO has higher separation efficiency and low recomb rate of photogenerated electrons and holes compared to TiO2.
Reply: Generally, the values of mh* and me* and the difference between them directly reflect the mobility of electrons and holes and the corresponding separation efficiency to some extent. In our work, the difference between the effective masses of electrons and holes of SrCu2O2 calculated is much larger than that of TiO2. Thus, we can make a conclusion that that SrCu2O2 is a semiconductor photocatalyst with higher separation efficiency and low recombination rate of photogenerated electrons and holes in comparison with TiO2. We have made a comment on page 4, line 147-148 in the manuscript.
Reviewer 2 Report
Journal: Materials (ISSN 1996-1944)
Manuscript ID: materials- 2180609
Type: Article
Title: Electronic, Optical, Mechanical, and Electronic Transport Properties of SrCu2O2: A First-principles Study.
Authors: Sheng Jiang, Dianhui Wang, Yan Zhong, Chengying Tang, Chaohao Hu *.
a) Keywords: Write 5 words.
b) Introduction: Write the objective of the present work carefully.
c) Why the authors didn’t measure the transmittance of the sample under testing for the optical properties part?
d) Fig. 6 x-axis legend put Energy photon (eV).
e) Why the authors didn’t measure the other mechanical properties of the samples such as wear, impact, surface roughness, hardness vigers, thermal conductivity, Flexural strength….etc.?
f) For references, choose recent refs. Please, refer to these refs. are very useful for the different measurement characterization
DOI: https://doi.org/10.1016/j.optmat.2019.109328
DOI: https://doi.org/10.1088/1742-6596/1795/1/012059
Best Regards
Author Response
Thanks very much for your suggestions about our manuscript. We have made revisions to the manuscript according to the comments, and point-to-point replies are listed as follows:
- a) Keywords: Write 5 words.
Reply: We have added the keyword "mechanical properties".
- b) Introduction: Write the objective of the present work carefully.
Reply: According the suggestion of reviewer, we have added a more in-depth description of the necessity and feasibility of the research about SrCu2O2 in the introduction. These can be seen on page 1, line 23-31, line 35-48 and page 2, line 69-71 of the revised manuscript.
- c) Why the authors didn’t measure the transmittance of the sample under testing for the optical properties part?
- e) Why the authors didn’t measure the other mechanical properties of the samples such as wear, impact, surface roughness, hardness vigers, thermal conductivity, Flexural strength….etc.?
Reply: Experimental studies can indeed improve the readability of the article. However, the present work is a theoretical study based on first-principles calculations. In this work, we have theoretically investigated the optical properties including complex refractive index, extinction coefficient, absorption coefficient, reflection rate, energy loss spectrum and mechanical properties including elastic constants, bulk modulus, shear modulus, Young’s modulus, Poisson’s ratio of SrCu2O2 in details, and also made a reliable comparison with some obtained experimental data. In addition, according to the best of our knowledge, the mechanical properties of samples like wear, impact, surface roughness are dynamic data and the effect of time should be considered. The static first-principles calculations may be not suitable for considering these problems. While the classic molecular dynamic simulations is probably a good choice.
- d) Fig. 6 x-axis legend put Energy photon (eV).
Reply: “Photon energy” is indeed more accurate here. We have modified the legend of Fig. 5 and Fig.6.
- f) For references, choose recent refs. Please, refer to these refs. are very useful for the different measurement characterization
DOI: https://doi.org/10.1016/j.optmat.2019.109328
DOI: https://doi.org/10.1088/1742-6596/1795/1/012059
Reply: The two articles are of great reference value to this work, and we have cited them in the revised manuscript (Ref. 3 and 4).
Reviewer 3 Report
In this work, the authors reported their ab-initio calculation results on properties of SrCu2O2, which includes crystal structure, lattice dynamics, band structure, optical properties, and electronic transport properties. The study is thorough and cover important fundamental properties of SrCu2O2, and this will contribute to development of transparent conducting oxide technology and oxide catalysts. I suggest the manuscript to be minor revised before considered for publication, and the minor revision would help to generate better interest and contribution in oxide technology. Following are my comments.
Comments:
1. I would suggest the authors to summarize the experimental results, so far, on the properties of SrCu2O2 related to this work. Therefore, it would provide better backgroud of this work and raise more interest from the readers.
2. I would suggest the authors simply write about effective mass instead of mobilityof carriers in line 129 on page 4 because the authors discussed only effective mass here and presented the mobility matirx later in the manuscript.
3. For the discussion of dielectric function, refractive index, and absorption coefficient on page 6, I would suggest the authors to add inserts for the spectral range of 1.5 ~ 3.3 eV for clearer comparison in the visible light regime.
4. The authors suspected the absorption peak at around 2.66 eV in their results coming from exciton absorption. When comparing to the band gap of 3.2 eV in their results, it seems to indicate an exciton binding energy up to almost 600 meV, which is extremely large. Can the authors clarify what kind of exciton level might it be or is there any other exciton absorption peak at an energy closer to band gap value?
5. The carrier concentration value in line 217 on page 8 looks strange, the authors shall check and correct it.
Author Response
Thanks very much for your suggestions about our manuscript. We have made revisions to the manuscript according to the comments, and point-to-point replies are listed as follows:
- I would suggest the authors to summarize the experimental results, so far, on the properties of SrCu2O2 related to this work. Therefore, it would provide better background of this work and raise more interest from the readers.
Reply: It indeed raises more interest from the readers to add experimental data on the related properties of this study. However, the current research on SrCu2O2 is relatively scarce. To provide better background of this work and raise more interest from the readers, we rewrote the introduction in the revised manuscript. The available experimental data of SrCu2O2 such as lattice parameters, band gap, and measured optical properties have been listed in the manuscript for comparison.
- I would suggest the authors simply write about effective mass instead of mobility of carriers in line 129 on page 4 because the authors discussed only effective mass here and presented the mobility matirx later in the manuscript.
Reply: We highly agree with this point of view, which can make our article more logical. The corresponding modification on page 4, line 138-139 in the revised manuscript has been made.
- For the discussion of dielectric function, refractive index, and absorption coefficient on page 6, I would suggest the authors to add inserts for the spectral range of 1.5 ~ 3.3 eV for clearer comparison in the visible light regime.
Reply: This is a good suggestion. In the revised manuscript, we have added the insets for better considering the data in the visible light area (1.5 ~ 3.3 eV) in Figure 5 and Figure 6 as suggested by the reviewers.
- The authors suspected the absorption peak at around 2.66 eV in their results coming from exciton absorption. When comparing to the band gap of 3.2 eV in their results, it seems to indicate an exciton binding energy up to almost 600 meV, which is extremely large. Can the authors clarify what kind of exciton level might it be or is there any other exciton absorption peak at an energy closer to band gap value?
Reply: We have checked carefully in the relevant article on excitons, and find that the binding energy of excitons in bulk semiconductors is generally around tens of meV. We are sorry for the lax writing. This may indeed be a problem written before. After re-analyzing the calculated data of the absorption coefficient, we believe that the value at 2.66 eV is not obvious compared to other peaks, and have made corresponding corrections in the revised manuscript.
- The carrier concentration value in line 217 on page 8 looks strange, the authors shall check and correct it.
Reply: The value of carrier concentration has been corrected in our revised manuscript.